SOFTWARE

# LCMSpector: A simple open-source viewer for targeted hyphenated mass spectrometry analysis

**Mateusz Fido**[1], **Etienne Hoesli**[2], **Elisa Cappio Barazzone**[3], **Renato Zenobi**[1], **Emma Slack**[3,4,5]*

**1** Department of Chemistry and Applied Biosciences, Swiss Federal Institute of Technology, Zürich, Switzerland, **2** Department of Biology, Swiss Federal Institute of Technology, Zürich, Switzerland, **3** Department of Health Sciences and Technology, Swiss Federal Institute of Technology, Zürich, Switzerland, **4** Sir William Dunn School of Pathology, University of Oxford, Oxford, United Kingdom, **5** Basel Research Center for Child Health, Basel, Switzerland

* emma.slack@hest.ethz.ch

## Abstract

The ubiquitousness of hyphenated mass spectrometry techniques across life sciences has made researchers around the world aware of their analytical power. However, the analysis of multidimensional high-complexity mass spectrometry data remains virtually impossible for non-specialists. LCMSpector is a standalone graphical user interface application for straightforward analysis of targeted mass spectrometry data, distributed under the MIT license and available free of charge for Windows, MacOS, and Linux. The user can upload and process the data entirely locally on their machine by specifying the desired mass-to-charge ratios of the targeted compound ions and simply clicking "Process". It allows for the processing of multiple files simultaneously, freely modifying and exporting graphs in real time, and calculating compound concentrations based on calibration standards. We show the versatility and applicability of LCMSpector to different kinds of mass spectrometry data by analyzing a series of publicly available datasets from various samples recorded on instruments from multiple vendors and by different research teams.

## Author summary

Mass spectrometry coupled with chromatography is among the most powerful analytical techniques used in the life sciences today. However, the generated datasets are large and complex, requiring specialized software or custom scripts for analysis. Therefore, data viewing and treatment in mass spectrometry create a barrier to entry for non-specialists within the field. Open-source, user-friendly tools for inspecting raw data, enabling evaluation of both chromatographic and mass spectrometric information, are lacking. Here we present LCMSpector, an easy-to-use graphical application for visualizing and analyzing hyphenated mass

**Data availability statement:** LCMSpector is distributed under the MIT license. The source code and executables for Windows, MacOS, and Linux, as well as detailed guides and installation instructions can be found free of charge on GitHub (https://github.com/MateuszFido/LCMSpector). As of the date of writing, LCMSpector is under constant development: the contents of this manuscript relate to the release version 0.9.10 (commit hash 95ad0e1, https://github.com/MateuszFido/LCMSpector/releases/tag/v0.9.10). Version 0.3 (script bundle) can be downloaded from Zenodo (DOI: https://doi.org/10.5281/zenodo.13990448). The overall test coverage of the codebase is 82% (source to test lines of code), with a 2:1 logic to UI testing ratio. The dependencies necessary for the installation and running of the non-executable versions are listed in the README and requirements text files. Documentation, learning resources and short articles describing the functionality are available on the project's wiki on GitHub: https://github.com/MateuszFido/LCMSpector/wiki. Links to the publicly available datasets reused in this article:—NIST mds2-2601 https://doi.org/10.18434/mds2-2601—license: public domain - MassIVE MSV000088442 https://doi.org/10.25345/C5M58V—license: CC0 1.0 - Public repository of ETH Zürich https://doi.org/10.3929/ethz-b-000663653—license (as stated by the repository host): https://rights-statements.org/page/InC-NC/1.0/—MassIVE MSV000092750 https://doi.org/10.25345/C5JD4Q057—license: CC0 1.0—MassIVE MSV000092751 https://doi.org/10.25345/C5DN40600—license: CC0 1.0.

**Funding:** ES, EH, and ECB were funded by the Basel Research Center for Child Health as part of the Multi-Investigator Project: Microbiota engineering for child health. ES acknowledges the support of the Swiss National Science Foundation (40B2-0_180953, 310030_185128), and European Research Council Consolidator Grant (865730). This work was supported as a part of NCCR Microbiomes, a National Centre of Competence in Research, funded by the Swiss National Science Foundation (grant number 180575) and by the LOOP Zurich mTORUS project (Helmut Horten Foundation). ECB was paid by the Basel Research Center for Child Health. MF, RZ, and ES received salaries from the ETH Zurich. EH did not receive any specific funding for this project. The funders had no role in study

spectrometry data, available on Windows, MacOS, and Linux. Using several public datasets, we demonstrate how LCMSpector's comprehensive raw data viewing features aid targeted analysis of metabolic products. We benchmarked our application against software with similar functionality and conducted a small-scale usability study to evaluate adoption by non-specialists. The results indicate that LCMSpector lowers the technical barrier of entry and improves efficiency for researchers analyzing hyphenated mass spectrometry data. We hope this open-source software benefits the life science community and grows into a robust tool across subfields, such as metabolomics, proteomics, glycomics, and other "omics" employing hyphenated mass spectrometry.

## Introduction

Hyphenated mass spectrometry (MS) techniques have revolutionized the way analytical sciences shape today's progress in science, industry, and the clinic [1–7]. Owing to the rapid development in instrumentation, spectrometers are now proficient at analyzing thousands of samples in record times and are capable of resolving many previously challenging isomeric and isobaric interferences. Liquid chromatography–mass spectrometry (LC–MS) has been at the forefront of analytical chemistry, becoming a leading technique and the analytical method of choice in drug discovery [1,8,9], pharmacology [10], the omics sciences [5,11,12] clinical sciences [7,13], toxicology [3,14,15] and many others [16–18]. Other branches of MS have made strides in the fields of environmental sciences [19], structural biology [20,21], imaging [22,23], forensics [24,25], and virtually any other area where the sample's structure, identity, composition, or amount are of value.

Nevertheless, one of the consequences of the high throughput, precision, and accuracy of high-resolution MS measurements is that they tend to generate enormous datasets in the process. This "big data" problem requires mass spectrometrists or their collaborators to become skilled at conversion, processing, interpretation, and integration of this complex data or to employ additional bioinformaticians and data analysts for this purpose. The closed-source nature of the MS vendors' application programming interfaces exacerbates this problem by making raw data inaccessible for processing without the usage of their proprietary tools. This leads to customer lock-in and promotes unhealthy dependency on the vendors' products and workflow monopoly, a regrettably widespread practice [26].

There have been several attempts at bridging the gap between generating highly complex MS data and their processing and interpretation. Initiatives such as ProteoWizard [27], XCMS [28], or mzio's MZmine [29] have brought together large communities of applied MS users and are now widespread among research groups and companies alike. Since their inception, many related open-source chemometric tools [30,31], programming language libraries [32,33], frameworks and applications [34,35] have been developed, further increasing the diversity of applied conversion, processing, and analysis methods.

design, data collection and analysis, decision to publish, or preparation of the manuscript. The authors declare no financial conflict of interest.

**Competing interests:** The authors have declared that no competing interests exist.

Despite this, many challenges remain in the field of MS data analysis. The multitude of instruments, possible data acquisition methods, workflows, and statistical methods used in life sciences make it unrealistic to apply a cut-and-dried analysis pipeline offered by most currently available software. Moreover, as MS data generally requires significant computational power, many analytical platforms require the user to upload their data onto an external server, further limiting their independence and shifting the trust to a third party, as the data is no longer processed locally on their workstation or personal computer.

Here, we present LCMSpector, which is a completely free-of-charge, fully local, and open-source program that provides a graphical user interface developed for viewing and analysis of MS data. It integrates several Python libraries, enabling data visualization processing of different MS and LC data formats, and simple quantitative analysis. In this work, we present the main features of LCMSpector and demonstrate its application to the preprocessing and analysis of several hyphenated MS data sets.

## Design and implementation

### Application structure

The main graphical user interface is implemented via the model-view-controller architecture (MVC) using the PySide6 framework and Python bindings for Qt (LGPL license) [36]. The main thread runs the user interface, while most of the resource-heavy computation is relegated to separate processes and threads using Python's built-in multiprocessing and concurrent.futures modules. This minimizes UI-thread blocking, maintains user interactivity, and makes maximum use of the available system resources. Files can be uploaded into LCMSpector via the "Upload" tab of the main tab view Qt widget, with a custom list view supporting both drag-and-drop functionality and manual file browsing by displaying the OS-native file browsing dialog. The absolute paths are stored as a list of strings and converted into derivatives of the Measurement class upon starting the processing. The subclasses of Measurement define several methods for preprocessing the raw data, storing the processed data as properties, and enabling additional plotting modules.

### Supported input formats

The application accepts two types of data: text files (TXT and CSV file formats) containing chromatography data in a column-wise format, and either MS files (mzML) or peaklists with pre-assigned retention times, also in text format. For text-based formats, LCMSpector automatically determines the delimiter and applies a custom algorithm, skipping any optional file header and looking for two columns containing floating-point numbers. The default behavior considers the first found column as retention time as the last column as the detector signal. mzML is an XML-based, vendor-agnostic file format that can be translated from any major manufacturer's propriety MS raw data file via tools such as MS Convert (ProteoWizard package) [27]. This allows for unified data sharing and analysis irrespective of the instrument and has been known for many years in the MS community alongside other formats such as mzXML, mzMLb, or HDF5 [37,38].

## Data processing

After loading the data, LCMSpector performs the initial preprocessing. If operating in the LC/GC–MS mode, the chromatography and MS data are processed concurrently. The chromatography data is parsed from any text file format containing columns of data convertible to Python floating-point numbers. Subsequently, baseline interpolation and correction are performed (described in the Results section). The program simultaneously reads the mzML files into memory, discarding any scans other than MS [1]. It then looks for the closest $m/z$ values to the user-specified ion lists within a specific mass-to-charge range (3 ppm by default). It then rebuilds the time-dependent ion traces (extracted ion chromatograms, XICs) for those $m/z$ ranges, optionally connecting them to their chromatography retention times and automatically annotating the run with the desired MS-identified compounds.

## Plotting

The plotting takes place in a separate module and is based on pyqtgraph, a Python scientific graphics and GUI Library for Python [39]. Pyqtgraph, as opposed to more popular and established libraries like matplotlib, maintains high user interactivity, enables real-time dynamic data processing such as log transformation, averaging, and fast Fourier transformation, and allows for exporting graphs in various formats. While not as mature as matplotlib (as of writing, 0.14.0 is the latest developer version of pyqtgraph), the library is comparatively lightweight, capable of performant plotting of millions of data points at the same time, dynamic panel docking and undocking, and embedding user-interactive and three-dimensional scientific graphs. It also provides limited capability for data export directly from the graph itself.

## Data export and further analysis

The export from LCMSpector is a CSV file containing column-grouped data consisting of the filename, the traced ion's name, its m/z value and an optional description, the ion's retention time, integrated MS and LC intensities, and optional concentration. The format facilitates further processing or graphing via table-oriented software (e.g., R, pandas, GraphPad Prism, OriginLab Origin, Microsoft Excel) by saving data in vertical (long) format. The output of every analysis was subsequently processed using R.

## Results

### Working principle

The easy-to-use interface of LCMSpector provides very simple access to the processing of raw MS data. The graphical layout of the application consists of a tab view, which is split into three sections: the Upload tab, where the user enters their data and provides initial parameters for the analysis. The Results tab, where they can view, modify, and export the graphs plotted for each provided file, and the Quantitation tab is used to create calibration curves and calculate concentrations based on relative ion counts (Fig 1).

Upon launching the application, the only available tab is the Upload tab, where the user can input their data. The main window area holds tables responsible for storing the absolute paths to MS data saved on the user's filesystem, including shared and network disks. Each table widget supports drag-and-drop functionality for easy access, on top of being able to browse the operating system's file manager with the "Browse" button. Every widget enforces proper input format by checking the extension of imported files—this means that the user can drag-and-drop entire folders into a widget and only the files compatible with it are going to be uploaded. There are several safeguards in place to ensure proper application state before starting processing – the user gets prompted with a warning in case an input file is corrupt or unreachable, or no ion list is selected for processing. The software also has several stages of logging implemented for debugging purposes, so that whenever an exception is caught, it bubbles up the callers to one of the main MVC modules (Fig A in S1 Text).

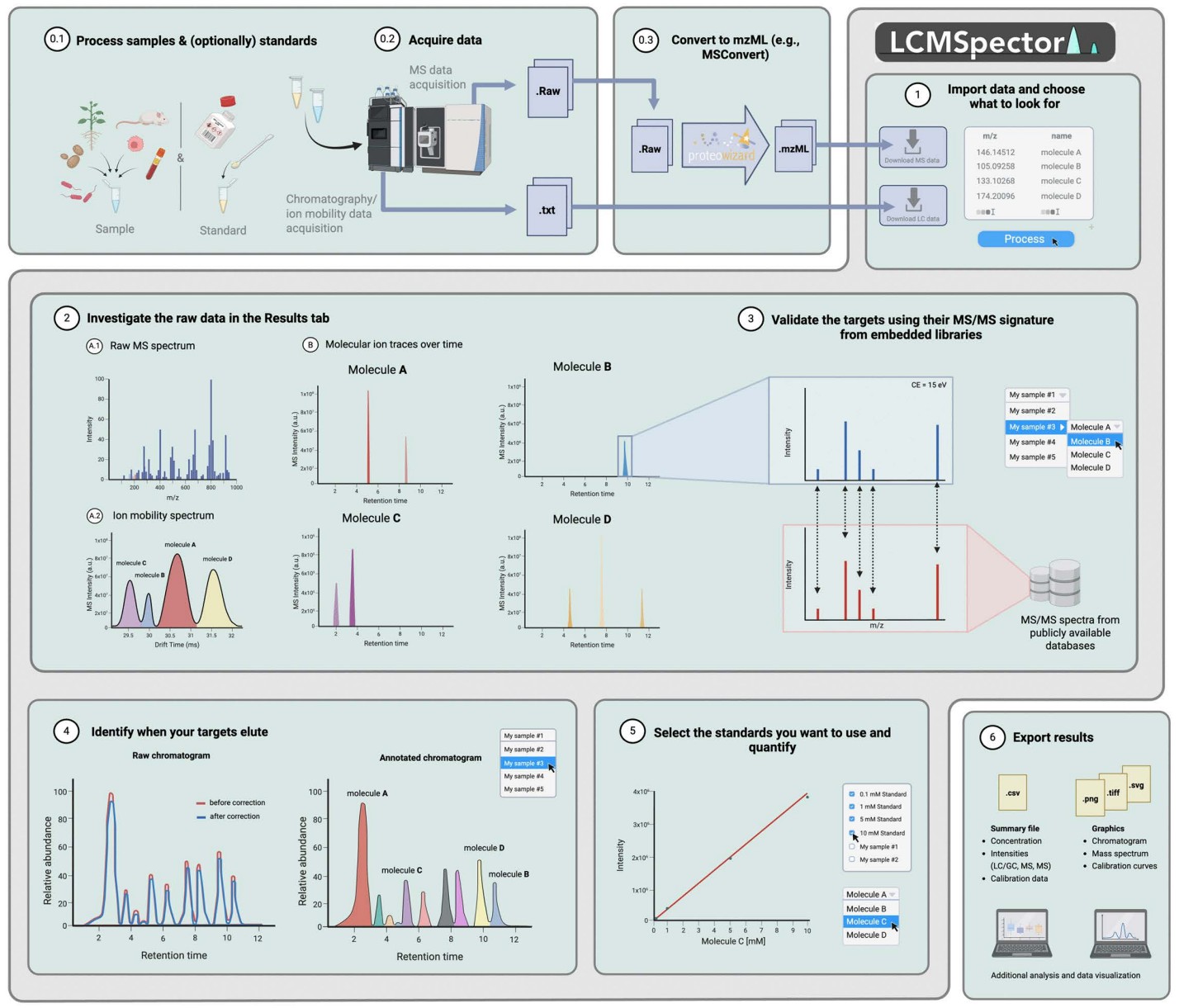

**Fig 1. Flowchart illustrating the data flow between components of LCMSpector's user interface and data processing architecture.** Created in BioRender with a publication license [40].

LCMSpector can work in three separate modes: "LC/GC–MS", "MS Only", and "Chromatography Only". The user can choose the desired mode by using the combo box in the top left-hand corner of the Upload tab. The application then switches the input type, depending on the mode. The LC/GC–MS mode requires data in text file formats for the chromatography data, and the open mzML file format for the MS data. It then uses those data to display the chromatogram, the corresponding mass spectra at all retention times within the chromatogram, the chromatogram annotated with the respective *m/z* values and extracted ion chromatograms on which the annotation is based. The MS Only mode only requires mzML files for processing, treating the total ion current as a chromatogram-like representation of the time-resolved signal.

The Chromatography Only mode requires the input of chromatography data, as well as retention time-based manual peak labels to display annotated data.

**Processing algorithms**

The core features of LCMSpector, aside from the graphical interface, lie in its preprocessing techniques. One of the main examples is its chromatography baseline correction, which utilizes the Statistical Non-linear Iterative Peak clipping algorithm [41], similarly to the hplc-py package [42], where we saw good results for even highly variable LC run backgrounds, including negative absorbance values. Briefly, the algorithm works in a series of steps:

1. Take the original chromatography signal (detector response over time), $S$

2. Compress the signal through approximation of the baseline, applying the compressing LLS operator [43]:

$$S_{LLS} = \ln\left[\ln\left(\sqrt{S+1}+1\right)+1\right],$$

where $S$ is the original chromatography signal. This step helps to reduce the impact of extreme values in the signal.

3. Perform iterative filtering to remove noise: The algorithm filters the compressed signal by selecting the minimum value between the current signal intensity and the average of the signal intensities at neighboring points:

$$S_{\text{iter}_i} = \min\left[S_{LLS_{i-1}}(t), \quad \frac{S_{LLS_{i-1}}(t-i) + S_{LLS_{i-1}}(t+i)}{2}\right],$$

where $i$ is the current iteration. This step helps to remove noise from the signal. LCMSpector uses a fixed default number of 20 iterations for all chromatography data, as we noticed this to be sufficient for most applications while keeping the computational preprocessing load relatively low.

4. Transform back: Finally, the filtered signal $S_{\text{iter}}$ is transformed back to its original scale with the inverse of the LLS operator and subtracted from the original signal (Fig 2).

Fig 2A demonstrates the application of the SNIP algorithm on a very noisy spectrum obtained from the GNPS-MassIVE repository (MSV000088442, Agilent 1260 HPLC and Bruker micrOTOF-Q MS) [44]. The interpolated background is visibly contributing to the signal as the run progresses. Upon subtraction, the resulting corrected UV spectrum contains much sharper peaks and an effectively null baseline. Fig 2B presents the performance of the algorithm on a GC spectrum recorded with an Agilent 8890/5977B GC–MS system (NIST Public Data Repository, https://doi.org/10.18434/mds2-2601) [45]. The plots are directly exported from LCMSpector (and recolored), facilitated by the underlying graphing library, which also allows extensive editing, removing, and transforming the plotted data in real-time.

Another very important algorithm in LCMSpector's processing pipeline is the construction of XICs, which is critical for proper MS-based compound quantification. It is also the most computationally expensive part of the processing step, needing to iterate over all the MS scans $S$ found in the input files. For each scan, it needs to process its $m/z$ array $M$, the corresponding intensities, and retrieve the relevant metadata, such as the injection time $T$. Then, for each compound $C$ targeted by the user, the ion current needs to be processed within the $m/z$ boundaries for the given ion $I$ expected for this compound, yielding MS peak area as a function of scan number (or time), otherwise known as the ion chromatogram.

Computationally, a naïve implementation would have a polynomial time complexity:

$$O(C \times I \times S \times (M + T))$$

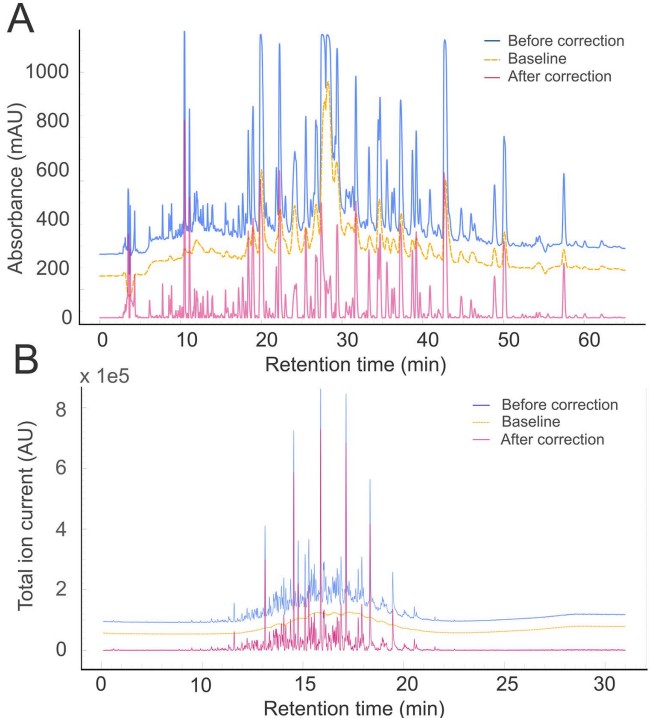

**Fig 2.** A) An LC chromatogram and B) a GC chromatogram before (blue, top) and after (red, bottom) applying the SNIP algorithm (20 iterations). The interpolated baseline is highlighted in yellow dashed lines (in between). Data taken from public data repositories A (GNPS-MassIVE repository, MSV000088442) and B (NIST Public Data Repository, https://doi.org/10.18434/mds2-2601).

where $T$ is a nested dictionary lookup, which in Python has a complexity of O(1). This therefore reduces to:

$$O(C \times I \times S \times M)$$

The current design of LCMSpector precomputes the scan metadata to avoid repeated lookups, groups ions avoiding re-scanning data for identical ion $m/z$ across compounds, and uses a binary search for $m/z$ array traversal, reducing per-scan work from linear $O(M)$ to logarithmic $O(\log M)$. The algorithm's complexity, while still mostly influenced by $C$, $I$, and $S$, is substantially faster in practice, especially as $M$ gets large for high-resolution instruments:

$$O(U \times S \times \log(M))$$

where $U$ is the unique ions grouped from all compounds ($U \leq C \times I$).

## Quantification

Targeted MS workflows often involve some kind of quantification, particularly within the omics sciences [46–48]. To support this, LCMSpector enables the user to select a subset of their input data as calibration data with known concentrations. Only one concentration value per file is supported, i.e., a single file represents a single calibrant data point, although an unlimited number of compounds is supported. Since this approach presupposes a large number of files, concentrations are imputed from the filename, though they can also be entered and modified manually. The user can choose to use either chromatography data or MS ion counts for the calibration procedure, depending on the mode. Upon selecting the

calibration files and clicking the "Calculate" button, a list of calibration curves is presented for every compound that was initially declared for tracing.

## Application to a biologically relevant dataset

The main purpose of LCMSpector is facilitating chromatographic and mass spectrometric investigation of biological samples, leveraging their analytical power without requiring an analytical background or programming skills. To illustrate this, we attempted to showcase the application's utility by examining publicly available LC–MS and GC–MS datasets acquired in a study of hospitalized patients with chronic liver disease, treated with the non-digestible disaccharide lactulose (GNPS-MassIVE repository IDs MSV000092750 and MSV000092751) [49]. The authors collected metagenomic and metabolomic data from fecal samples, providing insight into the expansion of commensal *Bifidobacteria* following lactulose treatment, reducing infections caused by antibiotic-resistant pathobionts, and resulting in improved patient outcomes. The main analytical focus of the work was bile acids, short-chain fatty acids, as well as tryptophan-pathway metabolites significant of microbial activity, such as indole derivatives. To verify and present the applicability of LCMSpector's processing pipeline to this type of a cross-sectional study, we performed quantitative analysis summarizing all the differentially abundant compounds from the cohorts presenting the expansion of *Bifidobacteria* (Fig 3A) and then performed additional exploratory work using LCMSpector's built-in and easy-to-use raw data viewing capabilities (Fig 3C–3E).

Fig 3B shows raw exports from LCMSpector's Quantitation tab. We recreated calibration curves for all the compounds for which the authors [49] recorded serial dilutions (bile acids and short-chain fatty acids). We subsequently used the calibration values to recalculate the concentrations of the relevant compounds in the fecal samples coming from healthy donors and liver disease patients (Fig 3C). A major advantage of LCMSpector is the easy graphical extraction of ion chromatograms alongside raw view data, allowing the user to recognize other features of the analysis that may increase quantitative accuracy. The data shown in Fig 3D shows an abundant potential dimer peak coeluting with the primary pseudomolecular ion for all the bile acids used for calibration (Fig 3D). The relative proportion of these species varied from sample to sample, in some cases having a higher abundance than the primary ion (Fig 3E). Gas-phase dimerization is a common non-covalent phenomenon at high analyte concentrations in electrospray ionization mass spectrometry [50,51]. LCMSpector easily allows for accurate hyphenated MS data analysis by enabling such untargeted discovery thanks to its raw data viewing capabilities.

## Non-standard mass spectrometry

Considering the continuous rapid evolution and development of MS, new hyphenated techniques emerge in the field on a regular basis. As a means of standardizing the input, LCMSpector accepts mzML as its file format of choice, as it is ubiquitous in the field, and its conversion is supported by all the major vendors [38]. This means that any MS data that can be written in mzML can be analyzed, including flow injection and direct infusion techniques.

One example of such a method is secondary electrospray ionization mass spectrometry (SESI-MS), wherein a gaseous analyte collides with a charged electrospray, producing secondary ions [52]. The method has gained popularity in the field of breath analysis and general analysis of trace vapors. Fig 4 shows the application of LCMSpector to SESI-MS data (ThermoFisher Scientific Q Exactive Plus) by using the MS Only mode. As the technique does not feature any chromatographic separation, the chromatogram is replaced by the total ion current, which serves as a global overview of the evolution of the signal across the time dimension.

As of today, there is no agreed-upon general analysis pipeline for the preprocessing and presentation of SESI-MS data. Most researchers default to analyzing and presenting data in the form of untargeted metabolomics, rarely investigating individual putative metabolites. LCMSpector was created with targeted approaches in mind, facilitating the exploration of small molecules underlying the examined experimental conditions, preferably confirmed with further measurements (e.g., with tandem MS, derivatization, standard spiking, etc.). Our hope is that LCMSpector allows diverse fields of researchers

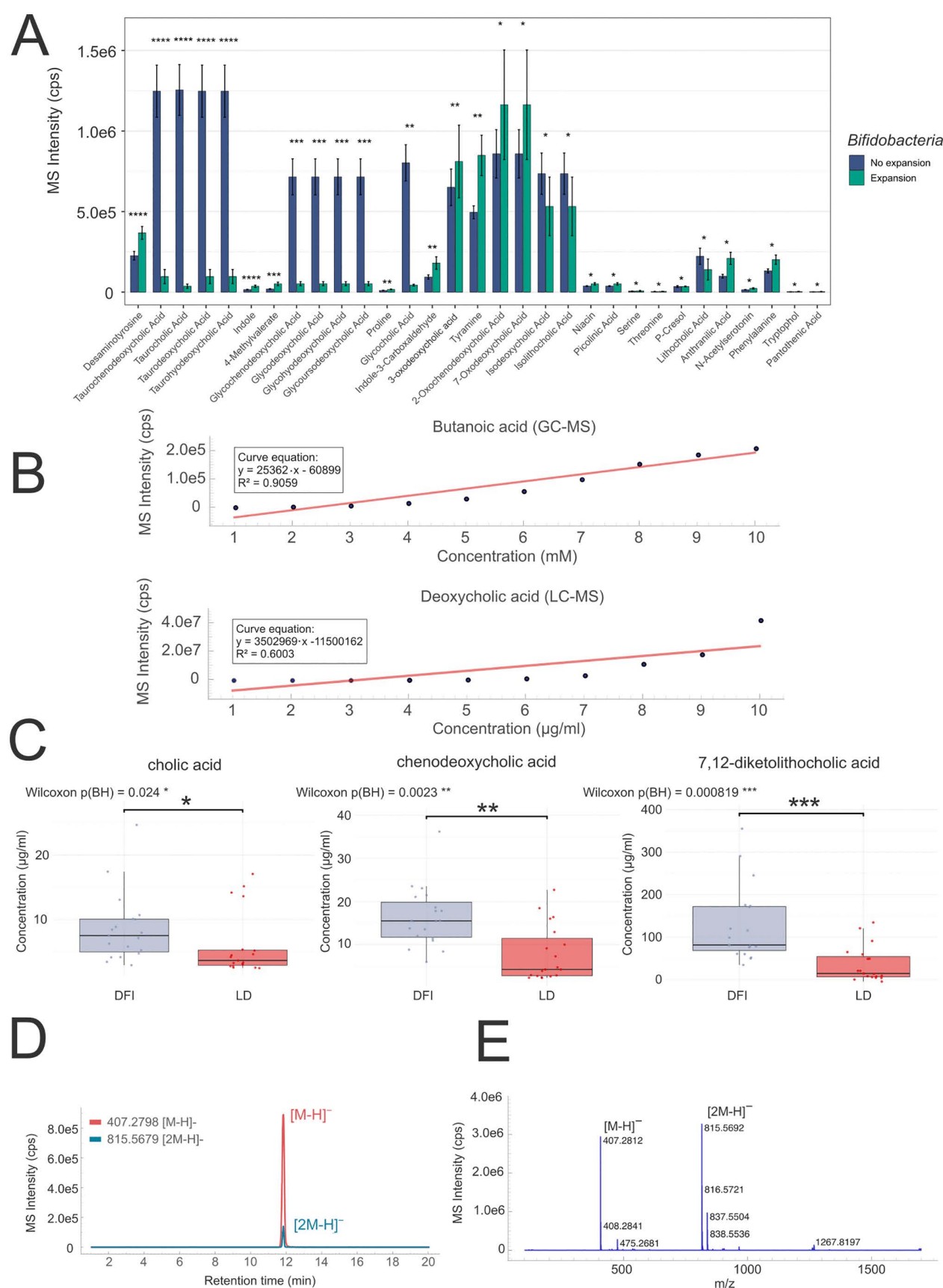

**Fig 3. Overview of the reanalyzed dataset from the study by Odenwald and colleagues (*Nat. Microbiol.* 2023, 8 (11), 2033–2049, MassIVE repositories MSV000092750 and MSV000092751) [49]. A)** Bar plots summarizing the most significant differentially abundant compounds in patients with greater than or equal to (green) and lower than (blue) 10% of total abundance of *Bifidobacteria* in fecal samples. **B)** Example calibration curves exported from LCMSpector based on the GC–MS (top) data for butanoic acid and LC–MS (bottom) data for deoxycholic acid. **C)** Example box plots of differentially abundant bile acids in the healthy (DFI, blue) and liver disease (LD, red) patient populations. **D)** Extracted ion chromatogram from a 10 µg/ml concentration standard solution of allocholic acid and **E)** mass spectrum showing the presence of a pseudomolecular dimer peak [2M − H]⁻ which could be included to further improve the accuracy of quantification of total intensity of allocholic acid.

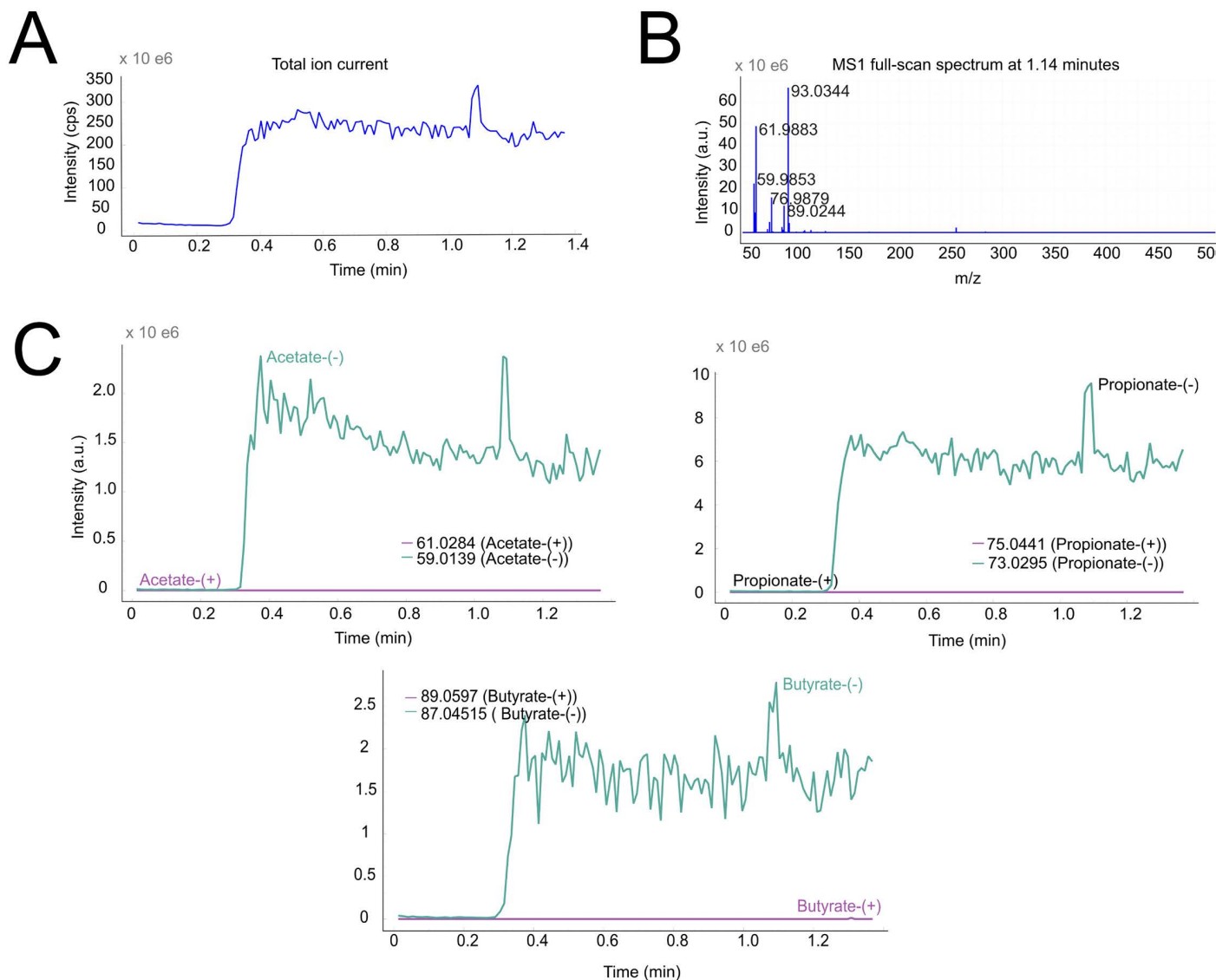

**Fig 4. SESI-HRMS samples of human breath gas, which were stored in an ethylene vinyl alcohol copolymer sampling bag produced in-house, direct graphical output of LCMSpector, part of a published dataset [53,54]. A)** Interactive plot of the total ion current, which displays the underlying mass spectrum **B)** when the user clicks on a corresponding scan time of the TIC. **C)** Extracted ion chromatograms of short-chain fatty acids present in the exhaled breath samples (acetic acid, propionic acid, butyric acid) can be seen correlating with the total ion current.

expand into targeted workflows and pursue confirmation of the traced metabolites, which our software aims to assist and expedite.

## Performance benchmarking

MS output is often classified as "big data" because of its high dimensionality, complexity, and sampling density. For example, LC–MS workflows produce a time-series signal at regular intervals for every ion they measure. With standard injection times in the order of tens to hundreds of milliseconds, an average MS run yields thousands of scans (e.g., 4,500 for a 15-min injection assuming 5 scans per second). Every scan carries multidimensional information consisting of the m/z values of detected ions and their intensities, and usually additional metadata such as charge states, polarity, retention time, etc. For high-resolution instruments (mass accuracy of 0.001 Da and greater) measuring between 100 and 1,000 *m/z*, the uncompressed intensity and m/z arrays would sum up to:

$$2 \cdot \frac{m/z_{high} - m/z_{low}}{\text{mass accuracy}} = 2 \cdot \frac{1000 - 100}{0.001} = 1.8 \times 10^6 \text{ data points}$$

per each scan, resulting in file size in the order of gigabytes (1 GB $= 8 \times 10^9$ bits).

Therefore, proper loading and in-memory handling of data can be considered the most important bottleneck for any MS software. To assess the performance of LCMSpector in these tasks, we conducted thorough scalability tests on all the major platforms (Windows, MacOS, and Linux), consisting of loading and processing a series of MS files converted to the mzML format. We benchmarked against the latest versions of two established open-source alternatives with very similar functionality: mzio's MZMine and OpenMS TOPPView (Fig 5).

Both LCMSpector and MZMine provide verbose logging functionality which allowed for exact measurement of the loading and processing time, whereas TOPPView's loading time was measured manually: from when files were dropped into the user interface until the UI became available for interaction, indicating loading was complete. Additionally, processing time for MZMine and LCMSpector was measured by performing targeted feature extraction of 40 *m/z* values—pseudomolecular ions for the 20 essential amino acids in both positive and negative ion modes. In TOPPView this operation is only possible on one layer (file) at a time and would otherwise necessitate the use of the FeatureFinder tool orchestrated via the TOPPAS workflow design GUI, and so extends beyond TOPPView's functionality. After completing the loading and processing steps (or just loading in the case of TOPPView), peak memory usage was assessed using the respective platform's resource monitor.

We found LCMSpector and MZMine to perform roughly on par in terms of loading and processing speed, both using parallelized multithreaded loading pipelines, with MZMine additionally incorporating memory-mapping for even faster file access. TOPPView was much slower on every platform, caused primarily by sequential I/O operations. In terms of resource usage, LCMSpector performed the best out of all three programs, using about half the memory of TOPPView, and around 2–4 times less than MZMine, depending on the platform. This is likely because only the essential scan data (no extra metadata) is loaded and smaller C data structures are used by compiling Python to C code with a source-to-source compiler, Nuitka [55].

We also assessed LCMSpector's peak integration and quantification by measuring precision and accuracy of its concentration estimates against known compound concentrations and comparing results to the vendor's algorithm (Thermo Scientific's Chromeleon) and another open-source alternative, MZMine. Five amino acids (aspartic acid, glutamic acid, histidine, lysine, serine) were combined into a 10 mM calibration mix and serially diluted to 1 mM, 100 µM, 10 µM, and 1 µM. Each dilution was directly infused into a high-resolution ThermoScientific Exploris 240 mass spectrometer and full-MS data were recorded to construct calibration curves. For all programs, we extracted XICs using the same m/z window: [M + H]$^+$ ± 0.0003 Da, integrated the peak intensities, and built 5-point linear calibration curves. We then interpolated

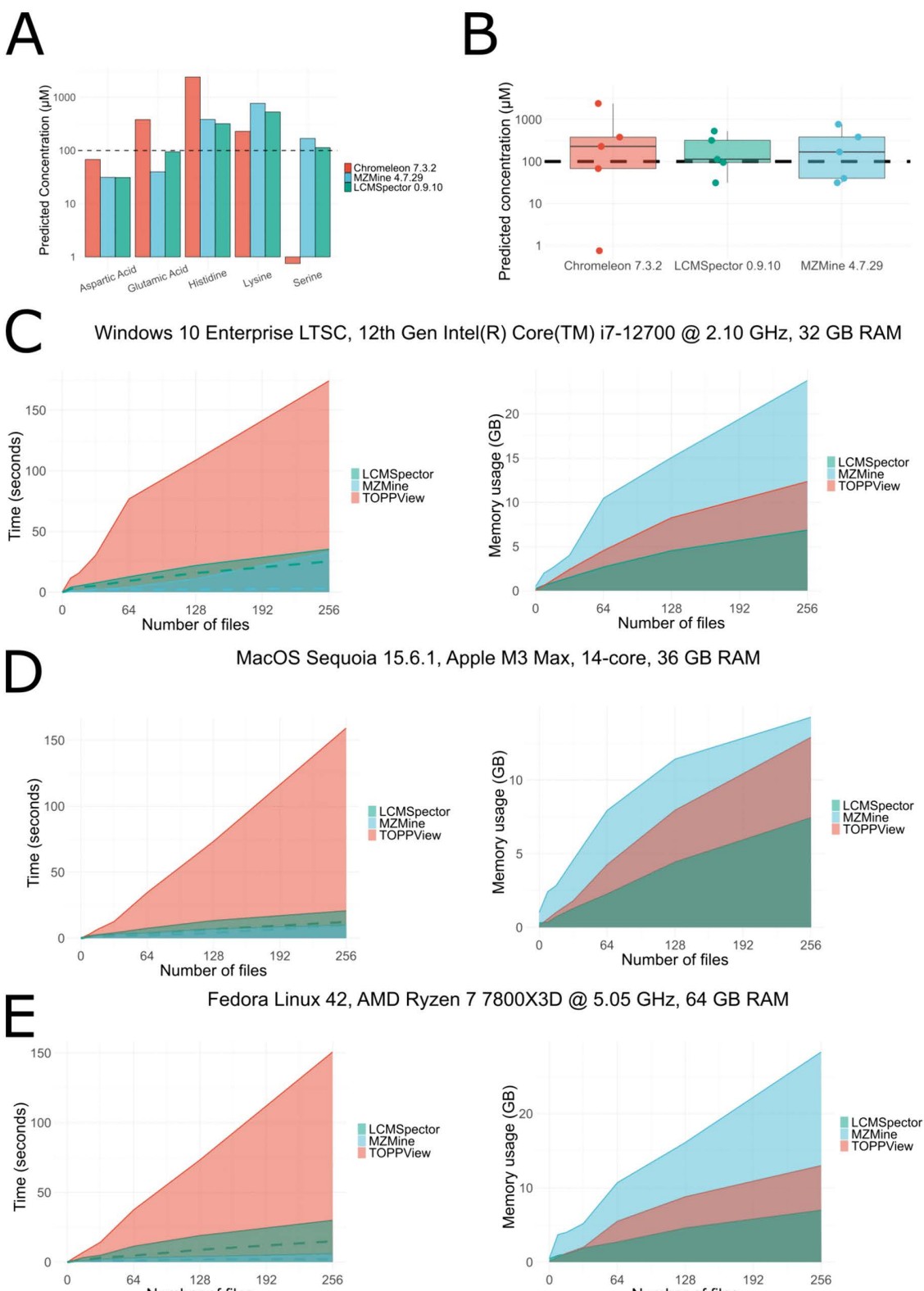

**Fig 5. Performance and quantitation capability benchmarking of LCMSpector.** Benchmarking data was generated on three different platforms, as denoted in each respective Fig's title. **A)** Barplots showing concentrations (logarithmic scale) of five different amino acids predicted by Chromeleon 7.3.2 (red, left), MZMine 4.7.29 (blue, middle), and LCMSpector 0.9.10 (green, right). The dashed horizontal line marks the real concentration (100 µM). **B)**

Residual boxplots showing the precision and accuracy of each software's prediction of a 100 μM concentration of each of the five tested amino acids. The dashed horizontal line marks the real concentration (100 μM). The solid horizontal line on each boxplot signifies the mean predicted concentration for all five amino acids. **C** Windows, **D)** MacOS, **E)** Linux performance comparisons of LCMSpector 0.9.10, MZMine 4.7.29 and OpenMS TOPPView 3.4.1 on. Left: Time taken to load (solid line) and process (dashed line) as a function of the number of input mzML files. Right: Peak memory usage as a function of the number of input mzML files. Red: TOPPView, blue: MZMine, green: LCMSpector.

each curve to estimate the concentration at the 100 μM point and compared those predicted values across Chromeleon, MZMine, and LCMSpector.

Somewhat surprisingly, we found the vendor algorithm (Chromeleon) to perform the poorest out of all three, whereas both LCMSpector and MZMine generally produced very accurate estimates of the real concentration for all 5 compounds. Based on our experience, this is most likely because Chromeleon's integration seems to be designed to work on well-defined, chromatographically-resolved peaks—its algorithms require a clear baseline and peak contours to work properly. LCMSpector also uses retention time-based peak integration when available, automatically defining peak boundaries and baseline-correcting the MS signal. However, if this strategy fails, there is a built-in fallback mechanism to sum the intensities for the given ion over time using trapezoidal integration, similarly to MZMine [56]. Nonetheless, LCMSpector's approach was the most precise and the most accurate among all three when tested on this data (Fig 4B).

### User-friendliness

To assess the approachability of LCMSpector to newcomers, we conducted a small usability study among ten users of different platforms that the application is distributed on (Windows, MacOS, and Linux). The users were given a set of instructions (Appendix A in S1 Text), consisting of downloading and installing the suitable executable for their platform, and then testing its various features on a set of sample LC-MS data. They were then asked to complete a short questionnaire on ease of use and functionality, rating installation difficulty, diversity of predefined compound lists, plotting-window operability, and reporting any bugs or confusing points (Appendix B in S1 Text). None of the users had previous experience in using software for MS data analysis.

The most represented platform was Windows (Fig 6). The median score for the easiness-of-use was 5 out of 6, indicating that no user found the application to be difficult or confusing. Most users described the installation process as straightforward.

Importantly, most basic functions, such as loading a file, starting the processing step, exporting the results and where the export is saved, were evident to all users. This indicates that the user-interface of LCMSpector was designed properly, and the core functionality is easily accessible and understandable.

With regards to potential improvements, the users cited stability, with some experiencing crashes when executing multiple functionalities at the same time, such as changing viewed files in the Results tab and reprocessing the data. This is a valuable indicator that additional safeguards against race conditions and potential premature garbage collection should be implemented, as well as expanding the codebase's integration testing as it matures and new features and functions are introduced.

### Limitations

Despite its performance benefits and user-friendliness, LCMSpector is not without limitations. Currently, the main barrier to widespread adoption is the lack of functionality supporting many features useful in mass spectrometry—customizable peak-picking algorithms for discovery (untargeted) workflows, better support for data-independent and single/multiple-reaction-monitoring experiments or more sophisticated ion mobility features (e.g., loading and viewing are currently possible, but 2D analysis is not). Likewise, one important pain point in community adoption is the handling of

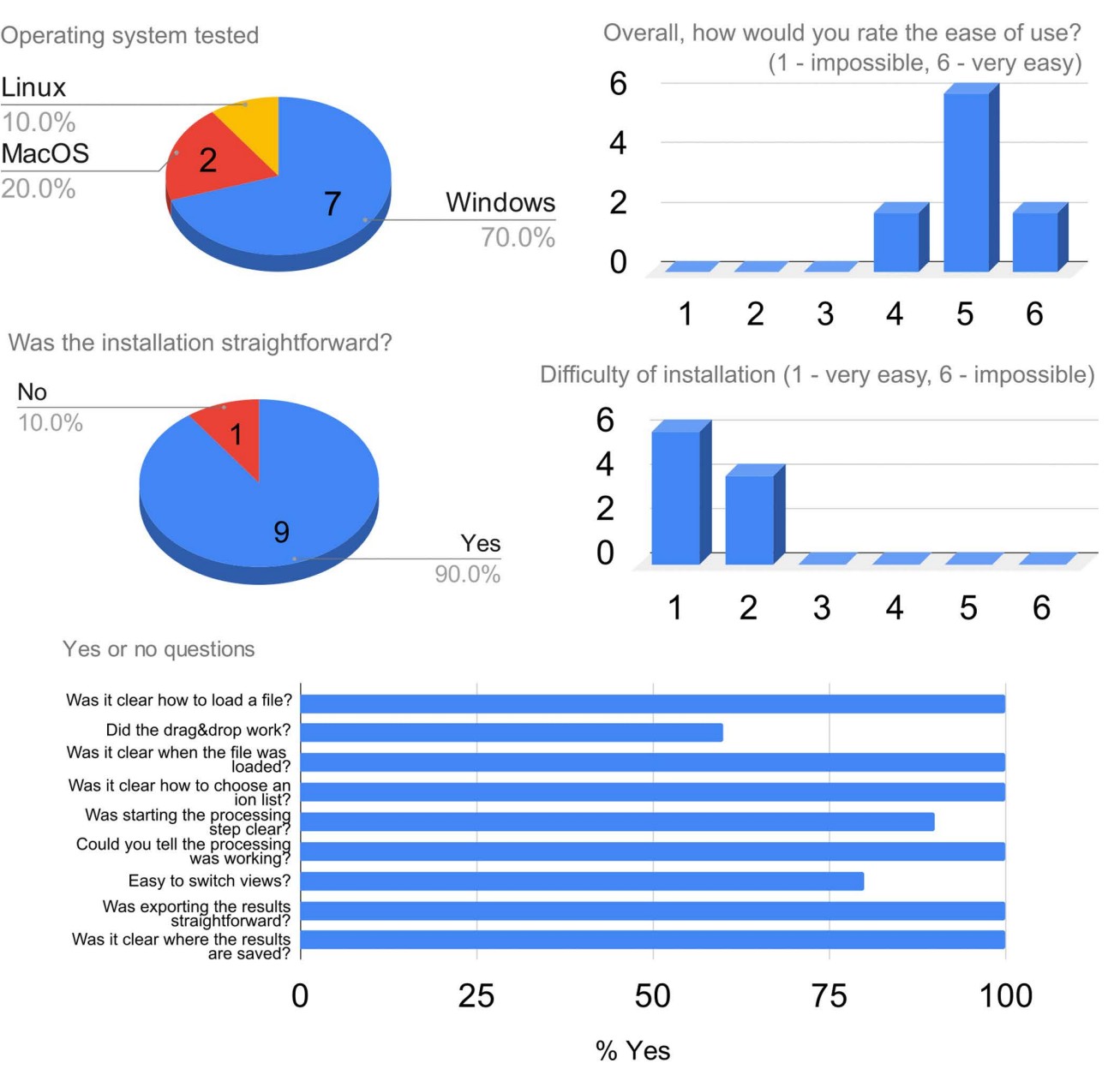

**Fig 6. Results of the usability study.**

proprietary file formats and other open file formats, not only mzML. Loading performance could also be improved upon by creating Cythonized versions of the pyteomics library's reader objects and incorporating memory-mapping, as in this regard LCMSpector is currently still outperformed by MZMine (Fig 5C–5E). Additionally, stability issues experienced by users of some platforms need to be carefully inspected to provide a fluid experience regardless of operating system and processor architecture. Nonetheless, all these points are included in the roadmap for LCMSpector's future development (Fig B in S1 Text) and we hope that with further development and community contributions, these barriers can soon be overcome.

## Availability and future directions

We presented the free, local, and open-source graphical interface software LCMSpector, enabling quick and simple viewing and investigation of MS data and discussed it in the scope of currently available technology. As MS becomes more widespread and common in laboratories throughout the world, parsing and evaluating the information it produces need to keep up with the technique's prominence. In our view, this means removing barriers of entry wherever possible and making the analysis streamlined and available also for non-experts, which is why we hope this software helps adopt targeted workflows with minimal experience in MS and data analysis. This could have broad appeal in the related fields, such as proteomics, lipidomics, glycomics, and metabolomics.

LCMSpector is distributed under the MIT license. The source code and executables for Windows, MacOS, and Linux, as well as detailed guides and installation instructions can be found free of charge on GitHub (https://github.com/Mateusz-Fido/LCMSpector). As of the date of writing, LCMSpector is under constant development: the contents of this manuscript relate to the release version 0.9.10 (commit hash 95ad0e1, https://github.com/MateuszFido/LCMSpector/releases/tag/v0.9.10). Version 0.3 (script bundle) can be downloaded from Zenodo (DOI: https://doi.org/10.5281/zenodo.13990448). The overall test coverage of the codebase is 82% (source to test lines of code), with a 2:1 logic to UI testing ratio. The dependencies necessary for the installation and running of the non-executable versions are listed in the README and requirements text files. Documentation, learning resources and short articles describing the functionality are available on the project's wiki on GitHub: https://github.com/MateuszFido/LCMSpector/wiki.

## Supporting information

**S1 Text.** Fig A. **(A)** The MVC architectural design of LCMSpector, created in BioRender with a publication license. Wetter, E. (2025) https://BioRender.com/4e4i8uvand. **(B)** LCMSpector data flow diagram. **Fig B.** Roadmap showing the plans for future development of LCMSpector. **Table A.** Windows—results of performance benchmarking between LCMSpector, MZMine and Sirius loading the same sets of mzML files. **Table B.** MacOS—results of performance benchmarking between LCMSpector, MZMine and Sirius loading the same sets of mzML files. **Table C.** Linux—results of performance benchmarking between LCMSpector, MZMine and Sirius loading the same sets of mzML files. **Fig D.** Calibration curve of aspartic acid from the calibration panel in Chromeleon Chromatography Studio (ver. 7.3.2) and calibration curve created based on the integrated intensities created in R (log–log). **Fig E.** Calibration curve of glutamic acid from the calibration panel in Chromeleon Chromatography Studio (ver. 7.3.2) and calibration curve created based on the integrated intensities created in R (log–log). **Fig F.** Calibration curve of histidine from the calibration panel in Chromeleon Chromatography Studio (ver. 7.3.2) and calibration curve created based on the integrated intensities created in R (log–log). **Fig G.** Calibration curve of lysine from the calibration panel in Chromeleon Chromatography Studio (ver. 7.3.2) and calibration curve created based on the integrated intensities created in R (log–log). **Fig H.** Calibration curve of serine from the calibration panel in Chromeleon Chromatography Studio (ver. 7.3.2) and calibration curve created based on the integrated intensities created in R (log–log). **Fig I** Calibration curve (log–log) of aspartic acid produced in R based on the integrated peak areas from MZmine version 4.7.28. **Fig J.** Calibration curve (log–log) of glutamic acid produced in R based on the integrated peak areas from MZmine version 4.7.28. **Fig K.** Calibration curve (log–log) of histidine produced in R based on the integrated peak areas from MZmine version 4.7.28. **Fig L.** Calibration curve (log–log) of lysine produced in R based on the integrated peak areas from MZmine version 4.7.28. **Fig M.** Calibration curve (log–log) of serine produced in R based on the integrated peak areas from MZmine version 4.7.28. **Fig N.** Calibration curve (log–log) of aspartic acid produced in R based on the integrated peak areas from LCMSpector 0.9.10. **Fig O.** Calibration curve (log–log) of glutamic acid produced in R based on the integrated peak areas from LCMSpector 0.9.10. **Fig P.** Calibration curve (log–log) of histidine produced in R based on the integrated peak areas from LCMSpector 0.9.10. **Fig Q.** Calibration curve (log–log) of lysine produced in R based on the integrated peak areas from LCMSpector 0.9.10. **Fig R.** Calibration curve (log–log) of serine produced in R

based on the integrated peak areas from LCMSpector 0.9.10. **Appendix A.** Instructions for the usability study of LCM-Spector. **Appendix B.** Contents of the usability study questionnaire.
(DOCX)

## Acknowledgments

The authors would like to thank the early-stage users of LCMSpector for helpful discussions and remarks when using the first versions of the software.

## Author contributions

**Conceptualization:** Mateusz Fido.

**Data curation:** Mateusz Fido, Etienne Hoesli.

**Formal analysis:** Mateusz Fido.

**Funding acquisition:** Renato Zenobi, Emma Slack.

**Investigation:** Mateusz Fido, Etienne Hoesli.

**Methodology:** Mateusz Fido.

**Project administration:** Mateusz Fido, Emma Slack.

**Resources:** Elisa Cappio Barazzone.

**Software:** Mateusz Fido.

**Supervision:** Renato Zenobi, Emma Slack.

**Visualization:** Mateusz Fido, Etienne Hoesli.

**Writing – original draft:** Mateusz Fido, Etienne Hoesli, Renato Zenobi, Emma Slack.

**Writing – review & editing:** Mateusz Fido.

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
