## [Decision Letter · Decision Letter 0]

15 Jun 2025

PCOMPBIOL-D-25-00816

LC-Inspector: a simple open-source viewer for targeted hyphenated mass spectrometry analysis

PLOS Computational Biology

Dear Dr. Slack,

Thank you for submitting your manuscript to PLOS Computational Biology. After careful consideration, we feel that it has merit but does not fully meet PLOS Computational Biology's publication criteria as it currently stands. Therefore, we invite you to submit a revised version of the manuscript that addresses the points raised during the review process.

Please submit your revised manuscript within 30 days (July 15, 2025, at 11:59 PM). If you will need more time than this to complete your revisions, please reply to this message or contact the journal office at ploscompbiol@plos.org. Please include the following items when submitting your revised manuscript:

We look forward to receiving your revised manuscript.

Kind regards,

Mohammad Sadegh Taghizadeh, Ph.D.

Academic Editor

PLOS Computational Biology

Shihua Zhang

Section Editor

PLOS Computational Biology

**Additional Editor Comments :**

Please note that the comments from Reviewer #1 has been attached.

**Journal Requirements:**

2) Your manuscript is missing the following sections: Design and Implementation, and Availability and Future Directions. Please ensure that your article adheres to the standard Software article layout and order of Abstract, Introduction, Design and Implementation, Results, and Availability and Future Directions. For details on what each section should contain, see our Software article guidelines:

https://journals.plos.org/ploscompbiol/s/submission-guidelines#loc-software-submissions

4) We notice that you stated "LC-Inspector is distributed under the MIT license. The source code and executables for Windows and MacOS, as well as installation instructions, can be found free of charge on GitHub (https://github.com/MateuszFido/LC-Inspector). Version 0.3 (designed to be run on a high-performance computing cluster) can be downloaded from Zenodo (DOI: https://doi.org/10.5281/zenodo.13990448)" under the heading "SUPPORTING INFORMATION." Please note that supplementary information should not be included in the manuscript file. It should be uploaded separately with the file type 'Supporting Information'. If this is not supplementary information, please amend the heading accordingly.

Potential Copyright Issues:

i) Figure 1a.  We note that the figure is created through BioRender. Please confirm that you hold a Premium account and provide a pdf copy of the CC BY 4.0 Licence as provided by BioRender. For instructions on how to generate a CC BY 4.0 license for your figure, please see the guidelines here: https://help.biorender.com/hc/en-gb/articles/21282341238045-Publishing-in-open-access-resources.

If you are using the free assets from BioRender, we are unable to publish these images as they are licenced under a stricter licence than CC BY 4.0. In this case we ask you to remove the BioRender images and replace them with open source alternatives.

See these open source resources you may use to replace images / clip-art:

- https://bioart.niaid.nih.gov/

- https://bioicons.com/

- https://healthicons.org/

- https://scidraw.io/

- https://reactome.org/icon-lib

- https://www.phylopic.org/images

6) Thank you for stating "All relevant data used to present applications of LC-Inspector are accessible from publicly available repositories and linked throughout the manuscript." Please note that your Data Availability Statement is currently missing the repository name, and the DOI/accession number of each dataset OR a direct link to access each dataset. Please update your Data Availability Statement to include this information.

7) Please amend your detailed Financial Disclosure statement. This is published with the article. It must therefore be completed in full sentences and contain the exact wording you wish to be published.

3) If any authors received a salary from any of your funders, please state which authors and which funders.

8) Please ensure that the funders and grant numbers match between the Financial Disclosure field and the Funding Information tab in your submission form. Note that the funders must be provided in the same order in both places as well. Currently, "the LOOP Zurich mTORUS project" and "the Stiftung für Naturwissenschaftliche und Technische Forschung" are missing from the Funding Information tab.

**Reviewers' comments:**

Reviewer's Responses to Questions

**Comments to the Authors:**

**Please note that one of the reviews is uploaded as an attachment.**

Reviewer #1: The authors did a great job on this manuscript LC-Inspector: a simple open-source viewer for targeted hyphenated mass spectrometry analysis to PLOS Computational Biology. The development of LC-Inspector is a commendable contribution to the mass spectrometry community, particularly in providing an accessible, user-friendly, and open-source tool for non-experts in the field. I have carefully reviewed your manuscript and would like to provide some constructive feedback to help improve its clarity and overall impact.

Reviewer #2: Fido et al. present LC-Inspector, which seems to be a useful utility for hyphenated MS data analysis. The manuscript is fairly straightforward in describing the software and its capabilities. I think the work is sound and I have no major critique. My rather minor comments below:

p2:32 (and elsewhere)

The abbreviation MS has already been introduced, so use it consistently instead of ‘mass spectrometry’.

p2:39

“vendor’s” should probably be “vendors’” because plural genitive.

p3:55

“LC-Inspector is a completely free-of-charge…” Note that LC-Inspector has not been introduced yet. Consider “Here, we present LC-Inspector, which is a completely free-of-charge…”

p3:64

“...concurrency and futures modules.” I suspect the authors mean “…concurrency.future modules”. Unless there coincidentally is a separate futures module that I cannot find.

p3:72-47

Text or csv files can have their data stored in many ways. If certain column orders or similar are expected, please mention that.

p4:85

“It then rebuilds the extracted ion chromatograms for those m/z ranges…” Maybe it is a convention I am unfamiliar with, or I misunderstand something, but is the term ‘ion chromatogram’ used for ions in a mass spectrometer? If so, fine, but if not, I suggest referring to it by another name, like a reduced mass spectrum or something. Especially since chromatography data is also part of the context. If I misunderstand, at least consider that the text can be misinterpreted as it is currently written.

p4:96

“comma-separated values (CSV)” CSV has already been discussed, so no point in introducing the abbreviation here.

p7:145

It would be prudent to cite the original paper Hampton et al. (https://doi.org/10.1016/0168-9002(94)91657-8) in relation to the LLS operator.

Lastly, I am curious about the name of the software. LC suggests liquid chromatography, but the central technique here is MS. Is there some history behind the name?

**Have the authors made all data and (if applicable) computational code underlying the findings in their manuscript fully available?**

Reviewer #1: Yes

Reviewer #2: Yes

PLOS authors have the option to publish the peer review history of their article (what does this mean? ). If published, this will include your full peer review and any attached files.

**Do you want your identity to be public for this peer review?** For information about this choice, including consent withdrawal, please see our Privacy Policy .

Reviewer #1: **Yes: ** Shafiu A. Umar Shinge

Reviewer #2: No

**Figure resubmission:**
---

## [Decision Letter · Decision Letter 1]

12 Oct 2025

PCOMPBIOL-D-25-00816R1

LCMSpector: a simple open-source viewer for targeted hyphenated mass spectrometry analysis

PLOS Computational Biology

Dear Dr. Slack,

Thank you for submitting your manuscript to PLOS Computational Biology. After careful consideration, we feel that it has merit but does not fully meet PLOS Computational Biology's publication criteria as it currently stands. Therefore, we invite you to submit a revised version of the manuscript that addresses the points raised during the review process.

Please submit your revised manuscript within 30 days (November 09, 2025, at 23:59). If you will need more time than this to complete your revisions, please reply to this message or contact the journal office at ploscompbiol@plos.org. Please include the following items when submitting your revised manuscript:

We look forward to receiving your revised manuscript.

Kind regards,

Mohammad Sadegh Taghizadeh, Ph.D.

Academic Editor

PLOS Computational Biology

Shihua Zhang

Section Editor

PLOS Computational Biology

**Journal Requirements:**

1) Please provide an Author Summary. This should appear in your manuscript between the Abstract (if applicable) and the Introduction, and should be 150-200 words long. The aim should be to make your findings accessible to a wide audience that includes both scientists and non-scientists. Sample summaries can be found on our website under Submission Guidelines:

2) Some material included in your submission may be copyrighted. According to PLOSu2019s copyright policy, authors who use figures or other material (e.g., graphics, clipart, maps) from another author or copyright holder must demonstrate or obtain permission to publish this material under the Creative Commons Attribution 4.0 International (CC BY 4.0) License used by PLOS journals. Please closely review the details of PLOSu2019s copyright requirements here: PLOS Licenses and Copyright. If you need to request permissions from a copyright holder, you may use PLOS's Copyright Content Permission form.

Potential Copyright Issues:

i) The following Figures contain screenshots: 1B, 1C, and Appendix A. We are not permitted to publish these under our CC-BY 4.0 license, websites are usually intellectual property and are copyrighted.This includes peripheral graphics of the web browser such as icons and button. We ask that you please remove or replace it.

3)  Please ensure that the files are uploaded in the online submission form in a correct numerical order

4) Please ensure that the funders and grant numbers match between the Financial Disclosure field and the Funding Information tab in your submission form. Note that the funders must be provided in the same order in both places as well. Currently, the Financial Disclosure states there was no funding received.

**Reviewers' comments:**

Reviewer's Responses to Questions

**Comments to the Authors:**

Reviewer #1: Dear Authors,

I have carefully examined your point-by-point responses and the revised manuscript. I am pleased to note that you have been exceptionally responsive and have undertaken a comprehensive revision that has substantially elevated the quality, rigor, and impact of your work. The additions and clarifications have effectively mitigated the core concerns I have raised in the initial review.

Specifically, the following revisions are particularly noteworthy and have enhanced the manuscript from a promising software description into a convincing and validated resource for the community

1. Introduction of rigorous performance benchmarks, the authors addition of head-to-head performance comparisons with established tools like MZmine and OpenMS TOPPView (Figure 5C-E) is a critical improvement. By quantifying loading times, processing speed, and memory footprint across three operating systems, you have moved beyond qualitative claims of efficiency to provide concrete, empirical evidence. The finding that LCMSpector achieves competitive speed while consuming significantly less memory is a strong and distinct advantage that you have now convincingly demonstrated.

2. Validation of quantitative fidelity against a gold standard, the manuscript quantitative analysis comparing LCMSpector's concentration estimates against those from the vendor's own algorithm (Chromeleon) and MZmine (Figure 5A-B) is exactly the type of validation required to build trust in a new analytical tool. The results, which intriguingly show superior performance of the open-source tools in this specific direct-infusion context, provide a powerful argument for the accuracy and reliability of your software's core algorithms.

3. Demonstration of biological utility through a case study, the authors new analysis of a published liver disease dataset (Figure 3) successfully bridges the gap between technical capability and biological discovery. This no longer feels like a simple demonstration, but an actual research application. The insight regarding the potential impact of dimer peaks on bile acid quantification is a perfect example of how your software's integrated, raw-data-viewing capability can lead to more accurate and insightful analyses a key selling point that is now powerfully substantiated.

4. Empirical support for user-friendliness, conducting a formal usability study (Figure 6) was an excellent initiative that moves discussions of the GUI from anecdotal to evidence-based. The positive feedback from novice users provides robust, independent validation of your central thesis: that LCMSpector successfully lowers the barrier to entry for complex MS data analysis.

5. Enhanced scholarly transparency and reproducibility, the authors’ inclusion of a dedicated ‘Limitations’ section, a future development roadmap (Fig. S2), and the freezing of the code version with a specific commit hash and Zenodo DOI represent best practices in open-source software publication. This level of transparency not only builds immediate credibility but also thoughtfully sets community expectations and facilitates future collaboration and development.

6. Improved conceptual and algorithmic clarity, the restructuring of the SNIP algorithm explanation into a step-by-step process and the detailed computational analysis of the XIC construction algorithm greatly enhance the manuscript's educational value and clarify the novel contributions of your implementation, effectively distinguishing it from a mere repackaging of existing libraries.

Despite the revisions are excellent, two minor points could be considered

1. The response letter mentions increasing test coverage to ‘about 90%’. For maximum transparency, the exact test coverage percentage could be included in the "Availability" section or the GitHub repository.

2. The stability issues noted in the usability study (crashes during simultaneous operations) are acknowledged as a point for future improvement. While it's good they are noted, a brief sentence in the ‘Limitations’ section could further demonstrate transparency.

The authors have undertaken a rigorous and highly effective revision process. They have responded to all my comments and significantly enhanced the scientific rigor, clarity, and impact of their work through the addition of extensive new data and analyses. The manuscript now provides convincing evidence for the performance, accuracy, and user-friendliness of LCMSpector, effectively positioning it as a valuable new open-source tool for targeted mass spectrometry analysis.

The revisions have fully addressed all my major and minor concerns. The manuscript now meets the high standards expected for a software publication in PLOS Computational Biology. The manuscript is now ready to proceed further for the Editor-in-Chief’s final decision.

Reviewer #2: The authors have reacted and responded to the comments from both reviewers in an adequate and satisfying way. I see no reason not to accept for publication.

**Have the authors made all data and (if applicable) computational code underlying the findings in their manuscript fully available?**

Reviewer #1: Yes

Reviewer #2: Yes

PLOS authors have the option to publish the peer review history of their article (what does this mean? ). If published, this will include your full peer review and any attached files.

**Do you want your identity to be public for this peer review?** For information about this choice, including consent withdrawal, please see our Privacy Policy .

Reviewer #1: **Yes: ** SHAFIU A. UMAR SHINGE

Reviewer #2: No

**Figure resubmission:**

**Reproducibility:**



---

## [Decision Letter · Decision Letter 2]

28 Nov 2025

Dear Dr. Slack,

We are pleased to inform you that your manuscript 'LCMSpector: a simple open-source viewer for targeted hyphenated mass spectrometry analysis' has been provisionally accepted for publication in PLOS Computational Biology.

Best regards,

Mohammad Sadegh Taghizadeh, Ph.D.

Academic Editor

PLOS Computational Biology

Shihua Zhang

Section Editor

PLOS Computational Biology

Reviewer's Responses to Questions

**Comments to the Authors:**

Reviewer #1: I am pleased with the authors’ careful and thorough responses to my previous comments. The revisions made to the manuscript have fully addressed the two minor issues I raised.

Regarding test coverage, the authors have now included the exact test coverage percentage (82%) in the ‘Availability’ section. This provides maximum transparency as requested and is a welcome addition.

Regarding stability issues, the authors have amended the ‘Limitations’ section to explicitly mention the stability issues encountered by some users during the usability study. This demonstrates a commendable level of transparency about the current state of the software and its areas for future improvement.

The authors have also taken care to address other editorial points, such as figure formatting. The manuscript is now significantly strengthened, clearly written, and presents a valuable open-source tool for the community. I have no further concern. The manuscript is now highly rigorous and scientifically sound which make it fit for publication in PLOS Computational Biology. The work is now ready to go further for the Editor-in-Chief’s final decision in line with the remaining peer reviewers’ comments.

**Have the authors made all data and (if applicable) computational code underlying the findings in their manuscript fully available?**

Reviewer #1: Yes

PLOS authors have the option to publish the peer review history of their article (what does this mean? ). If published, this will include your full peer review and any attached files.

**Do you want your identity to be public for this peer review?** For information about this choice, including consent withdrawal, please see our Privacy Policy .

Reviewer #1: **Yes: ** Shafiu Adam Umar Shinge

---

## [Editor Report · Acceptance letter]

PCOMPBIOL-D-25-00816R2

LCMSpector: a simple open-source viewer for targeted hyphenated mass spectrometry analysis

Dear Dr Slack,

I am pleased to inform you that your manuscript has been formally accepted for publication in PLOS Computational Biology. Your manuscript is now with our production department and you will be notified of the publication date in due course.

With kind regards,

Anita Estes
